# Quantitative Measurement of Swallowing Performance Using Iowa Oral Performance Instrument: A Systematic Review and Meta-Analysis

**DOI:** 10.3390/biomedicines10092319

**Published:** 2022-09-18

**Authors:** Raffaella Franciotti, Erica Di Maria, Michele D’Attilio, Giuseppe Aprile, Federica Giulia Cosentino, Vittoria Perrotti

**Affiliations:** 1Department of Neuroscience, Imaging and Clinical Sciences, G. d’Annunzio University of Chieti-Pescara, 66100 Chieti, Italy; 2Department of Innovative Technologies in Medicine & Dentistry, G. d’Annunzio University of Chieti-Pescara, 66100 Chieti, Italy; 3Independent Researcher, 00165 Rome, Italy; 4Department of Medical, Oral and Biotechnological Sciences, G. d’Annunzio University of Chieti-Pescara, 66100 Chieti, Italy

**Keywords:** dysphagia, swallowing, tongue endurance, tongue pressure

## Abstract

Swallowing is a complex but stereotyped motor activity aimed at serving two vital purposes: alimentary function and the protection of upper airways. Therefore, any impairment of the swallowing act can represent a significant clinical and personal problem that needs an accurate diagnosis by means of reliable and non-invasive techniques. Thus, a systematic review and meta-analysis was performed to investigate the reliability of the Iowa Oral Pressure Instrument (IOPI) in distinguishing healthy controls (HC) from patients affected by swallowing disorders or pathologies and conditions that imply dysphagia. A comprehensive search was conducted following the Preferred Reporting Items for Systematic reviews and Meta-Analyses (PRISMA) guidelines and using PubMed, Scopus, Web of Science, Cochrane, and Lilacs databases. Overall, 271 articles were identified and, after a three-step screening, 33 case-control and interventional studies reporting IOPI measurements were included. The methodological quality of the retrieved studies resulted in being at a low risk of bias. The meta-analysis on case-control studies showed that maximum tongue pressure (MIP) values were always higher in HC than in patients, with an overall effect of the MIP difference of 18.2 KPa (17.7–18.7 KPa CI). This result was also confirmed when the sample was split into adults and children, although the MIP difference between HC and patients was greater in children than in adults (21.0 vs. 15.4 KPa in the MIP mean difference overall effect, respectively). Tongue endurance (TE) showed conflicting results among studies, with an overall effect among studies near zero (0.7 s, 0.2–1.1 s CI) and a slight tendency toward higher TE values in HC than in patients. Among the intervention studies, MIP values were higher after treatment than before, with a better outcome after the experimental tongue training exercise than traditional treatments (the MIP mean difference overall effect was 10.8 and 2.3 KPa, respectively). In conclusion, MIP values can be considered as a reliable measure of swallowing function in adults and in children, with a more marked MIP difference between HC and patients for the children population. MIP measures in patients are also able to detect the best outcome on the tongue function after the training exercise compared to traditional training.

## 1. Introduction

Swallowing may appear as a simple, obvious, effortless act, but it instead implies various complex mechanisms that involve all the levels of the central nervous system (CNS) as well as 25 pairs of muscles in the oropharynx, larynx, and esophagus. Indeed, the oral cavity, pharynx, and larynx—though anatomically separated—are functionally combined for the sequential motor responses that ensure chewing, speech, and swallowing itself. For these reasons, swallowing should be intended as a complex but stereotyped function that constitutes an essential motor activity in mammals together with an alimentary function that also aims to protect the upper airways, thus serving two vital purposes [1,2,3]. Therefore, any impairment of the swallowing act can represent a significant clinical as well as personal problem that should thus be properly and promptly detected and diagnosed.

Not by chance, it is possible to outline “dysphagia” as a subjective perception of difficulty during the passage of a liquid or solid bolus from the mouth to the stomach, or of obstruction during swallowing [4,5]. Dysphagia may occur in pediatric patients, with an incidence of 0.9% [6], due to many causes, such as structural abnormalities and prematurity, as well as neuromuscular, cardiopulmonary, gastrointestinal, and iatrogenic conditions [7]. Nevertheless, it is a very common condition that affects the elderly population, but also patients with Parkinson’s disease, dementia, stroke or brain injury, and head and neck cancer [8,9,10]. In particular, despite the incidence and prevalence of dysphagia across the spectrum of possible underlying conditions being not precisely known, it is suggested that 15% of older adults are affected by dysphagia [11]. It is essential to consider that dysphagia may entail various sequelae, impacting the individual’s quality of life, not only due to its physical effects—such as a lowered nutritional status and risk for food aspiration, pneumonia, and potential death—but also due to its severe social and psychological effects [12,13].

Notable effort has been made to assess swallowing, especially through quantitative measures, to distinguish physiological conditions from pathological or impaired ones. Among these, a videofluoroscopic swallow study (VFSS), also known as a modified barium swallow, is the present-day gold standard that ensures the evaluation of the swallowing function, consisting of a dynamic X-ray examination of the oral cavity, pharynx, and cervical esophagus through the employment of liquids and solids of different consistencies to assess swallowing fluoroscopically [14]. A fiberoptic endoscopic evaluation of swallowing (FEES) is another wide-employed diagnostic tool and uses a fiberoptic rhinopharyngoscope to provide detailed information about the anatomy and physiology of the pharyngeal stage of swallowing, also contributing to evaluate the presence, degree, and type of dysphagia [15,16]. When compared to VFSS, the reliability of this procedure for the identification of swallowing abnormalities was verified and both the above mentioned examination techniques resulted in managing dysphagic patients successfully [17,18].

Notwithstanding, many other techniques have been described and investigated as reliable assessment tools to study the swallowing act, especially in a valid, easy-to-perform, and non-invasive way, with a low level of discomfort for the patient, such as surface electromyography (sEMG) [19].

The Iowa Oral Performance Instrument (IOPI) is a handheld, portable pneumatic device with a light array that provides visual feedback of the pressure generated on an air-filled bulb. The bulb has an approximate length of 3.5 cm, a diameter of 4.5 cm, and an internal volume of 2.8 mL. The bulb is connected to an 11.5 cm long clear plastic tube (Figure 1a). Pressure is measured in kilopascals (kPa).

This innovative tool offers the possibility to effectively measure the pressures performed by the tongue on the bulb [20] after inserting the air-filled balloon inside the mouth and pressing the bulb against the roof of the mouth. The pressures measured are displayed on an Liquid Crystal Display (LCD) screen (Figure 1b,c).

Maximal isometric tongue pressure is measured by placing the bulb posterior to the upper incisors on the alveolar ridge (Figure 1b). Maximum isometric posterior tongue pressure is measured by placing the bulb 10 mm anterior to the most posterior circumvallate papilla (Figure 1c) [21]. To obtain the maximum tongue pressure (pMax), subjects are then asked to push the tongue towards the hard palate as hard as possible [22]. Tongue endurance is measured by placing the bulb in the desired location (anterior or posterior) and capturing the number of seconds (s) a subject can maintain tongue pressure at 50% of pMax. Visual feedback from the IOPI and verbal encouragement from the clinician are provided during the endurance trials. The trial ends when the pressure steeply drops or when 50% cannot be maintained for a few seconds [23]. In addition, tongue protrusion can be measured, with the holder positioned between the upper and lower incisors and the tongue bulb facing intraorally. Patients are then instructed to protrude the tongue as hard as possible against the bulb [24].

Since the tongue musculature changes the shape and the position of the tongue during swallowing [25], the evaluation of the function of tongue muscles—thus the measures of tongue pressures—lends itself well to the assessment of the swallowing function [26]. Moreover, IOPI provides a biofeedback for oral motor exercise and objectively quantifies patient performance. This systematic review was based on the clinical need to obtain objective measures and the pragmatic need for easy-to-use devices. The aims were to investigate the reliability of IOPI (1) to distinguish patients with swallowing disorders or conditions that imply dysphagia from healthy controls (HC), regardless of age (2), and (3) to determine the impact and effectiveness of tongue training exercises on swallowing performance compared to traditional training.

## 2. Materials and Methods

This systematic review was performed according to the guidelines of the Preferred Reporting Items for Systematic Reviews and Meta-analyses (PRISMA) statement [27]. The review record has been approved by the international prospective register of systematic reviews PROSPERO under the identification number CRD42022297506. The current review clearly addresses a focused question by using the participant, intervention, comparison, and outcomes (PICO) criteria.

### 2.1. Search Strategy

An electronic search on scientific databases (PubMed, Scopus, Web of Science, Cochrane, and Lilacs, ACM Digital, EBSCOhost, and Google Scholar) was performed to identify suitable studies, published after 31 December 1999, using the following terms and keywords alone or in combination: (“Iowa Oral Performance Instrument”) AND (swallow* OR deglut* OR tongue) AND (abnormal* OR normal* OR physiolog* OR typical OR atypical OR disorder OR dysfunction OR dysphagia) AND (assess* OR analy* OR evaluat* OR quanti* OR measure*) NOT (review OR “case report*” OR “case series” OR preclinical OR animal).

The first search was performed on December 2021. The last electronic search was performed up to April 2022, and updated on September 2022. In addition to the electronic search, reference lists of the selected studies were manually screened. A reference manager software program (Zotero, George Mason University, Fairfax, VA, USA) was used and the duplicates were discarded first electronically, and then by checking the resulting list manually.

### 2.2. Eligibility Criteria

All of the inclusion/exclusion criteria have been summarized in Table 1.

#### 2.2.1. Inclusion Criteria

The search was limited to studies published in the English language. The search was restricted to human studies with healthy or diseased children or adults of both genders included in the study population. Only studies where IOPI was used were selected. IOPI outcomes, such as tongue pressures and tongue endurance, had to be reported. Case-control and intervention studies were eligible to be part of this systematic review.

#### 2.2.2. Exclusion Criteria

All of the studies reporting any other diagnostic tool and not referring to IOPI measures were excluded [28,29,30]. In addition, studies published in journals without an impact factor and not peer-reviewed were eliminated [31,32,33,34,35,36,37]. When the mean and standard deviation of the outcomes were not shown in the articles, a request by email was sent to the corresponding authors and only the articles that provided the raw data about mean and standard deviation values were included [38,39,40]. Case-reports, preclinical studies, reviews, systematic reviews, and metanalyses were excluded.

### 2.3. Focused PICO Question

Participants: Patients with swallowing disorders or pathologies and conditions that imply dysphagia;Intervention: IOPI;Comparison: HC;

Outcomes: Maximum tongue pressure (MIP), lingual swallowing pressure (LSP), tongue endurance (TE), tongue protrusion pressure (TPS). We hypothesized that:IOPI is a reliable tool to distinguish HC from patients with swallowing disorders or pathologies and conditions that imply dysphagia.IOPI reliability is similar for children and adults.IOPI is able to measure an improvement in swallowing performance following traditional treatments and tongue training exercises in HC and in patients.

### 2.4. Selection of Studies

Retrieved citations were independently screened by two authors (EDM and FGC) and relevant studies were identified based on title and abstract. If those did not provide sufficient information about the inclusion criteria, the full text was evaluated to assess eligibility. Any disagreement was solved by discussion, and a third reviewer was consulted to make final decisions (VP).

### 2.5. Data Extraction and Analysis

Author and year, study design, duration of the study, participant baseline characteristics (number, mean age, age range, gender, pathologies), intervention, drop-out/lost to follow-up, follow-up duration, and outcomes (MIP, TE, LSP, and TPS) were extracted independently from each included study by two authors (EDM and FGC) using a predesigned data extraction form. Microsoft Excel 2020 (Microsoft Office, Microsoft Corporation, Redmond, WA, USA) was used for data collection and for descriptive analysis. A third reviewer was consulted when difficulties arose (VP). The primary outcomes included MIP. The secondary outcomes were TE, LSP, TPS.

### 2.6. Assessment of Methodological Quality

The quality assessment of the included studies was independently performed by two reviewers (EDM and FGC) as part of the data extraction procedure.

Specifically, the Newcastle–Ottawa Scale (NOS) [41] was applied for case-control and cross-sectional studies to judge each study on eight items distributed in three categories: “Selection”, “Comparability”, and “Exposure”. As each item corresponded to a multiple-choice question, depending on the answer, each study could be awarded with a maximum of one star for each item of the Selection and Exposure categories, whereas a maximum of two stars could be given for the Comparability category. Each study could be judged with a maximum of 9 stars.

ROBINS-I tool [42] was used for non-randomized intervention studies through “Bias due to confounding”, Bias in selection of participants into the study”, “Bias in classification of interventions”, “Bias due to deviations from intended interventions”, “Bias due to missing data”, “Bias in measurement of outcomes”, and “Bias in selection of the reported result” domains. Each study was assessed with one out of five possible overall judgements: low, moderate, serious, or critical risk of bias, or no information. RoB 2 tool [43] was used for randomized intervention studies instead through “Randomization process”, “Deviations from intended interventions“, “Missing outcome data”, “Measurement of the outcome”, and “Selection of the reported result”. Each study was assessed with one out of three possible overall judgements: low risk of bias, some concerns, or high risk of bias.

As it was practically unfeasible to keep patients and operators blinded to treatment, the related performance bias (blinding of participants and operators) was not accounted for, except for 4 studies [44,45,46,47] where conditions were explicitly blinded. Any disagreement was solved by discussion or consulting a third reviewer (VP) until consensus was achieved.

### 2.7. Statistical Analysis

Statistical analyses were conducted on the outcome variables of all selected studies that passed the eligibility criteria. To perform the statistical analyses, studies were divided into case-control and intervention studies. Spearman correlations and Student’s *t*-tests were carried out between outcome variables when applicable. Meta-analyses were performed if data on the outcome variables were provided in at least 4 studies. To answer the first PICO question, we selected all studies that measured at least one of the outcome variables in HC and patients. Due to the heterogeneity of the diseases, we included in the patients’ group all participants with a diagnosis related to swallowing disorders. To answer the second PICO question, the case-control studies were divided into two groups (children and adults) according to the age of the involved participants. To answer the third PICO question, we selected all studies that measured at least one of the outcome variables before and after traditional treatments and/or tongue training exercises. Age and sex of the population were heterogeneous among the studies. Thus, for the meta-analyses, we calculated the mean difference in each outcome variable between HC and patients as effect index for the case-control studies and the mean difference in each variable between pre- and post-treatment for the intervention studies. The weight of each study was performed according to the sample size and standard deviations of the outcome variables, as was performed previously [48]. For each effect index and for the overall effect, 95% confidence interval (CI) was estimated.

## 3. Results

### 3.1. Review Analysis

The search resulted in a total of 271 articles: 84 retrieved from PubMed, 23 from Cochrane, 2 from Lilacs, 77 from Web of Science, 85 from Scopus, 0 from ACM Digital, 81 from Google Scholar, and 32 from EBSCOhost. After duplicates being removed, 87 studies were available for the screening. An initial screening of the titles and abstracts identified 47 studies. After reading the full texts, 33 studies [44,45,46,47,49,50,51,52,53,54,55,56,57,58,59,60,61,62,63,64,65,66,67,68,69,70,71,72,73,74,75,76,77] were included, thus eliminating 14 studies [28,29,30,31,32,33,34,35,36,37,38,39,40,78] that did not meet the inclusion/exclusion criteria. The summary of the search strategy is depicted in Figure 2.

The reasons for study exclusions and characteristics of the included studies are presented in Table 2.

### 3.2. Characteristics of the Studies

#### 3.2.1. Case-Control Studies

Impaired swallowing performance was associated to many pathologies and conditions. In particular, among the 19 studies [49,50,51,52,53,54,55,56,57,58,59,70,71,72,73,74,75,76,77] with a case-control design, participants included patients with head and neck cancer in 3 studies (21.4%) [71,72,76], with muscular dystrophies in 4 studies (28.5%) [70,73,74,75], and dysphagic [77], post-extubated [49], unilateral cleft-lip-palate [50], Parkinson’s disease [51], chronic temporo-mandibular disorders (TMD) [52], sleep breathing disorders [53], and post-stroke [54] subjects in 1 study (7.1%), respectively. Among the 5 case-control studies with a cross-sectional design, older adults [55], patients with mouth breathing behavior [56], motor speech disorders and sound disorders [57], Sjogren syndrome [58], and Down syndrome [59] were included in only 1 study (20%).

#### 3.2.2. Intervention Studies

Among the 14 intervention studies [44,45,46,47,60,61,62,63,64,65,66,67,68,69], participants included dysphagic post-stroke subjects in 3 studies (21.4%) [45,46,60], older adults in 4 studies (28.6%) [47,61,62,63], and subjects with sleep breathing disorders in 4 studies (28.6%) [64,65,66,67]. Subjects with tongue thrust [68], as well as patients with Parkinson’s disease [69], were included in only 1 study (7.1%). Moreover, the sample in 1 out of 14 intervention studies (7.1%) was totally composed of HC without any reported pathologies [44]. In these 14 intervention studies [40,41,42,43,56,57,58,59,60,61,62,63,64,65], some sessions of tongue exercises were applied in order to evaluate the effect of oral musculature training on the swallowing performance. In particular, in 1 study (7.1%) [60], patients were treated with tongue-to-palate resistance training (TPRT), consisting of separately pressing the anterior and posterior region of the tongue strongly against the palate, and conventional dysphagia therapy, whereas the control group performed only conventional dysphagia therapy, including thermal tactile stimulation, facial massage, and various maneuvers. In 2 studies (14.2%) [46,63], participants performed tongue pressure and accuracy training (TPSAT), consisting of pressing both parts of the tongue against the palate as hard as possible, and generating precise pressures in either the anterior or posterior bulb location. In 1 study (7.1%) [65], the experimental group performed myofunctional therapy, consisting of isometric and isotonic oropharyngeal exercises divided into three categories: (1) nasal breathing rehabilitation, (2) labial seal and lip tone exercises, and (3) tongue posture exercises; in 1 study (7.1%) [45], the experimental group performed conventional dysphagia therapy and effortful swallowing training (EST), consisting of pushing the tongue onto the palate, while squeezing the neck muscles, and swallow as forcefully as possible, whereas the control group performed saliva swallowing without intentional force from the tongue and the neck muscles, and conventional dysphagia therapy, including orofacial muscle exercises, thermal tactile stimulation, expiratory training. In 2 studies (14.2%) [44,61], the experimental exercise was divided into isotonic and isometric types; in 1 study (7.1%) [62], participants performed a home-based mHealth app therapy program based on three therapy maneuvers: (1) effortful prolonged swallowing, (2) effortful pitch glide, and (3) effortful tongue rotation; in 1 (7.1%) study [68], participants underwent orofacial myofunctional therapy (OMT), consisting of isotonic and isometric exercises involving the tongue, soft palate, and lateral pharyngeal wall; in 3 studies (21.4%) [64,66,67], participants performed an AirwayGym app myofunctional therapy, based on exercises that are aimed at improving the tonicity of the various muscles involved in the pathogenesis of OSAHS; in 1 study (7.1%) [47], one group performed a tongue progressive resistance exercise (effortful water swallow) whereas the comparative group performed an instrumental tongue isometric exercise (lingual pressing tasks) using IOPI; in 1 study (7.1%) [69], participants in the experimental group performed traditional tongue exercises (tongue presses against the palate and lateral lingual movements) and instrumental tongue pressure exercises using the IOPI, whereas the comparative group performed only traditional tongue exercises.

All of the characteristics of the studies and IOPI outcomes are summarized in Table 3.

### 3.3. Evaluation of Methodological Quality

#### 3.3.1. Case-Control Studies

Among case control studies, a low risk of bias was found in eight studies [49,50,54,71,72,75,76,77]. Six studies instead [51,52,53,70,73,74] resulted in being at an intermediate risk of bias because of inadequate information about “Selection”, “Comparability”, and “Exposure” domains, according to the Newcastle–Ottawa quality assessment scale.

The Newcastle–Ottawa quality assessment scale was also applied to studies with a cross-sectional design: a low risk of bias was found in three studies [56,58,59]. Two studies instead [55,57] resulted in being at an intermediate risk of bias because of inadequate information about “Selection”, “Comparability”, and “Exposure” domains.

The quality assessment of case-control studies is available in Table 4.

#### 3.3.2. Intervention Studies

Among randomized intervention studies, a low risk of bias was found in eight studies [44,45,46,47,60,64,65,69]. One study instead [61] resulted in being at a moderate risk of bias because of inadequate information about the “Randomization process” domain, according to the RoB 2 tool (Figure 3).

A low risk of bias was also found in four out of five non-randomized studies [62,66,67,68]. Only one study [63] showed an overall moderate risk of bias, due to the “Bias in selection of participants into the study” domain (Table 5).

### 3.4. Statistical Results

For the case-control analyses, 21 [49,50,51,52,53,54,55,56,57,58,59,65,67,70,71,72,73,74,75,76,77] studies (63.64%) reported at least one of the outcome variables in the HC and patients’ groups. Specifically, 19 studies [49,50,51,53,54,55,56,57,59,65,67,70,71,72,73,74,75,76,77] (57.57%) reported MIP values, 4 studies [51,52,54,74] (12.12%) reported LSP values, 9 studies [36,40,45,61,62,63,64,66,67] (27.27%) reported TE values, and 2 studies [52,58] (6.06%) reported TPS values (Table 6). MIP values were obtained by pushing the tongue against the roof of the mouth as hard as possible, thus expressing tongue pressure. TE values were measured by asking the subjects to sustain 50% of their maximum pressure for as long as possible. Only 1 study [74] obtained TE by asking the subjects to hold 25% of their maximal pressure. TPS values were obtained by asking the subjects to protrude their tongue as hard as possible against the bulb, positioned between the upper and lower incisors and facing intraorally. LPS values were defined as the swallowing pressures generated swallowing boluses.

Due to the low number of studies, LSP and TPS measurements were excluded from all statistical evaluations. No significant correlation was found between MIP and TE values (*p* = 1). A meta-analysis showed that, in all studies, MIP values were always higher in HC than in patients (Figure 4), with an overall effect of 18.2 KPa (17.7–18.7 KPa CI).

Differently from MIP values, TE showed contrasting results among studies (Figure 5). Indeed, in three studies (9.09%) [71,72,74], TE values were lower in HC than in patients, in three other studies (9.09%) [50,54,77], TE values were similar for the two groups, and in the remaining three studies [59,73,76] (9.09%), TE values were higher in HC than in patients. The overall effect was near zero (0.7 s, 0.2–1.1 s), showing a slight prevalence of studies showing higher TE values in HC than in patients. However, the study [77] with the largest sample size (*n* = 150 in total) reported a TE mean difference between the two groups near the zero value, with the CI overlapping the y-axis (0.3 s, 1.8–2.3 s).

When the studies were divided according to participants’ age for the 19 studies [49,50,51,53,54,55,56,57,59,65,67,70,71,72,73,74,75,76,77] reporting MIP values, 7 studies (36.84%) [50,56,57,59,65,70,75] were related to children (7 < age < 18) and the remaining 12 studies (63.15%) [49,51,53,54,55,67,71,72,73,75,76,77] were related to adults with 40 < age < 73 years. One study [75] (5.26%) reported MIP values for different age ranges (7–8, 9–10, 11–12, 13–14 and 15–18 age), and thus all values were used in the meta-analysis related to children results (Figure 6a). The meta-analysis on children showed that, for only 1 study (5.26%) [75] (7–8 age range), the MIP value was lower in HC than in patients (−1.9 KPa, from −3.4 to −0.4 KPa). Thus, the overall effect evidenced higher values of MIP in HC than in patients (21.0 KPa, 14.6–27.5 KPa).

The meta-analysis on studies conducted on adults (Figure 6b) showed that, in all of the studies, MIP values were higher in HC than in patients, reporting an overall effect of 15.4 KPa (CI: 13.5–17.4 KPa). It should be noted that, in children, the difference between HC and patients was larger than in adults (21.0 vs. 15.4 in the overall effect). For TE values, only two studies [50,59] (6.06%) were related to children; thus, the meta-analysis was not performed for children and adults separately.

To assess if IOPI outcomes were able to highlight possible effects of treatments on swallowing performance, interventional studies on HC and patients were included. From all investigated studies and outcomes, 11 intervention studies [44,45,46,47,60,61,62,63,66,67,69] (33.33%) had MIP mean values before and after therapy and were used for statistical analyses. Eight studies [45,46,60,62,63,66,67,69] (24.24%) reported MIP mean values in patients before and after experimental training exercises (Table 7). Specifically, MIP values were significantly higher after experimental training exercises than before (*p* = 0.003 from paired t-test). In addition, the mean percentage change in MIP values before and after the tongue training exercise was 39.6%.

Only three studies [44,47,61] (9,09%) measured MIP values before and after the experimental training exercise in HC, and the mean percentage change was 25.1%. From these studies, two [44,61] (6.06%) measured MIP values twice also, but in the absence of intervention, and the mean percentage change between the two measurements was 1.7%.

Four studies [45,46,60,66,69] (12.12%) reported MIP values before and after traditional training in patients, and the mean percentage change was 8.7.

Figure 7 shows MIP values before and after treatments for all intervention studies in HC and in patients.

Meta-analyses showed that patients benefited from traditional trainings (Figure 8a), having MIP values greater after the treatment (the overall effect was 2.3 KPa, with 1.8–2.8 KPa of CI). The tongue training exercise (Figure 8b) had a better outcome compared to the traditional treatments, showing a higher overall effect among studies (10.8 KPa, 10.3–11.3 KPa CI).

## 4. Discussion

This systematic review showed the useful of IOPI as a quantitative measure of swallowing performance. Indeed, in the case-control studies, MIP values were always greater in HC than in patients. This result was confirmed by meta-analysis, suggesting that the MIP value is a quite reliable measure that may be used effectively by clinicians as a non-invasive measure of impaired swallowing performance related to different pathological conditions. Indeed, the studies included patients affected by sleep breathing disorders [53], head and neck cancer [71,72,76], muscular dystrophies [70,73,74,75], unilateral cleft-lip-palate [50], Parkinson’s disease [51], sleep breathing disorders [53], post-stroke and post-oral endotracheal extubation period [49,54], risk of malnutrition [55], mouth breathing behaviors [56,65], motor speech disorders and sound disorders [57], and Down syndrome [59], as well as oral phase dysphagia itself [77]. Therefore, according to these results, patients showed weaker tongue pressure—indicative of a scarcer function of tongue muscles due to the presence of any pathology or condition that might affect the swallowing performance—when compared to HC. Among all of the MIP measurements, the most evident difference between HC and patients was shown in the study by Rodrìguez-Alcalà et al. [67] for the OSAHS group, despite the limited sample size. IOPI MIP measures might also be used in the prodromal condition of dysphagia, which may represent a herald of disease progression [79].

On the contrary, the same might not be said for the analysis of TE measurements. There were studies [71,72,74] with lower TE values in HC than in patients, other studies [50,54,77] with similar TE values for the two groups, and, in the remaining studies [59,73,76], TE values were higher in HC than in patients. A slight prevalence of studies showing higher TE values in HC than patients was found. This was prevalently due to Rogus-Pulia et al., 2016 [76], who compared HC with patients with head and neck cancer and reported a very high difference (20.5 s, 23.9–27.2 s CI) between the two groups for the TE value, with a sample size similar to the other studies (*n* = 21, for each group). Since no clear pattern could be deduced from this analysis and a wide dissimilarity among all of the results was enhanced—probably due to the low number of studies investigating this measure—at the moment, TE cannot be considered as a reliable variable to distinguish healthy and diseased subjects, suggesting that further investigations of the above-mentioned measure should be pursued. Not by chance, the contrast between the acceptable reliability for tongue pressure and that of TE measurements was also noticed by Adams et al., 2013 [80]. Indeed, the results reported by this author showed—differently than tongue pressure values—TE mean values with above 10% changes between the first two consequent trials, despite the decrease in the subsequent trials; moreover, unacceptably large typical errors and weak-to-moderate intra-class correlation coefficients were found for TE measurements.

The meta-analysis on children showed higher values of MIP in HC than in patients, except for one study [75] conducted on subjects in a 7–8 age range; however, this was a slight difference (−1.9 KPa, −3.4 to −0.4 KPa) compared to all of the others [50,56,57,59,65,70,75]. A further analysis of the results showed that the difference between HC and patients was larger in children than in adults, thus suggesting that, among patients, children with pathological conditions have a worse tongue weakness than adult patients. In fact, children seem to show a less efficient activation of faster, higher-threshold motor units (type II) compared to adults, leading to a lower maximal pressure output [81]. Tongue pressure also changes with age in HC; in particular, a rapid increase has been noticed across ages 3–8, with a following slower increase until peaking in late adolescence to young adult age, according to Potter and Short [82]. This might explain the MIP results after dividing the studies according to participants’ age: the difference between HC and patients seems to be more evident in children, probably because diseased children show a weaker tongue not only due to the pathology, but also because of age itself and the ongoing changes in younger developing subjects.

The data analysis of the intervention studies showed a greater improvement in patients after experimental tongue training exercises, especially when a worse MIP value was found at baseline conditions [63,67]. Rodrgìguez-Alcalà et al. [67] showed the greatest difference in MIP values before and after training measurements. The results from Kim et al. [62] did not show a great improvement, probably because the only impaired condition taken into account in this study was defined as “complaints of swallowing difficulties (i.e., increased aspiration rate and foreign body sensation in throat)”, thus not referring to a particularly disabling comorbidity. Tongue pressure also showed improvements after traditional trainings in patients, even if it was definitely lower than after experimental trainings [45,46,60,69]. O’Connor-Reina et al. [66] also reported MIP values in patients with OSAHS that did not adhere to therapy. These MIP values were not considered in the meta-analysis because these patients were comparable to patients that did not receive any intervention. Indeed, the after-training tongue pressure values were almost similar to the before-training ones—even slightly lower—unlike adhering patients, who showed improvements after training. Not by chance, the adhering patients had worse baseline tongue pressures, unlike non-adhering patients; therefore, the better compliance could be due to greater motivation in the first place. This enhances how experimental tongue training, sometimes in association with traditional training [45,60,69], is effective at improving tongue function, especially in patients with more compromised baseline conditions. Traditional trainings might reveal itself as helpful at improving tongue function, but at a lower rate. The successful effects of experimental tongue training on patients might be important, as an improvement in tongue pressure might correspond to an improvement in the severity of the disease in some cases. For example, Suzuki et al., 2020 [83] observed significantly increased tongue pressures after myofunctional therapy (MFT), consisting of some exercises aimed at functionally obtaining the appropriate positioning of the tongue in OSA patients treated with continuous positive airway pressure (CPAP); a significantly decreased apnea–hypopnea index (AHI) was also found, suggesting that tongue exercise may have contributed to the improvement in the severity of OSA.

The tongue training exercise also showed improved tongue pressures in HC [44,47,61]. On the contrary, only a slight difference was found in HC in the absence of intervention [44,61]: the mean tongue pressure value after a period of no intervention was slightly higher than the baseline. This might mean that the periodical measuring of tongue pressure itself might lead to a greater activity of tongue muscularity and thus explain the time effect [84]. All of this might appear as non-useful information, but, according to Lin et al. [84], experimental tongue training in HC proposes instead the possibility of pressuring the tongue, giving rise to positive changes that can prevent or halt the progressively altered swallowing mechanism characteristic of healthy aging and hence representing a relevant strategy not only for dysphagia intervention but also for prevention. Finally, the meta-analysis on intervention studies confirmed that patients benefited from experimental training exercises, as revealed by the increase in MIP values after the therapy. Patients improved swallowing functions following traditional trainings, as tongue pressure was always greater after intervention, but at a slower rate.

## 5. Conclusions

IOPI MIP is a reliable measure of swallowing function in HC and in patients with different pathologies. MIP values were always higher in HC than in patients.

MIP values were also higher in HC than in patients among children, with a more remarkable difference than in adults.

Using TE instead cannot be considered as a reliable variable to distinguish healthy and diseased subjects, suggesting that further investigations are needed.

Experimental tongue training exercises alone or in combination with traditional trainings succeed at improving and pressuring the tongue function in patients, in an objectively better way than traditional trainings alone.

Therefore, IOPI proved itself to be a valid tool to successfully measure tongue pressure and detect the productive effects of tongue training on both healthy and diseased subjects.

## Figures and Tables

**Figure 1 biomedicines-10-02319-f001:**
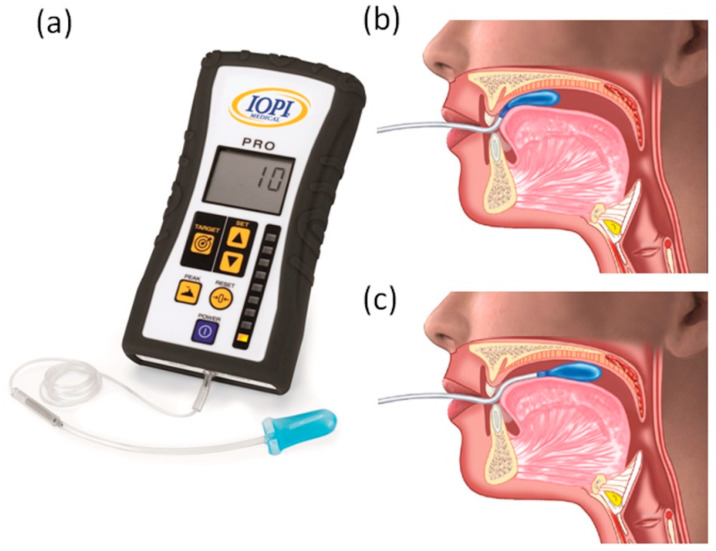
Iowa Oral Performance Instrument (IOPI). (**a**) An air-filled pliable plastic tongue bulb is connected through a clear plastic tube to the IOPI device, which displays the pressure measured. The tongue bulb can be positioned against the patient’s hard palate, just behind the alveolar ridge, (**b**) or at the transition between the hard and soft palate (**c**). The tongue compresses the bulb against the roof of the mouth, and the pressures exerted are measured and displayed on the IOPI LCD screen. Reproduced with permission from IOPI Medical LLC.

**Figure 2 biomedicines-10-02319-f002:**
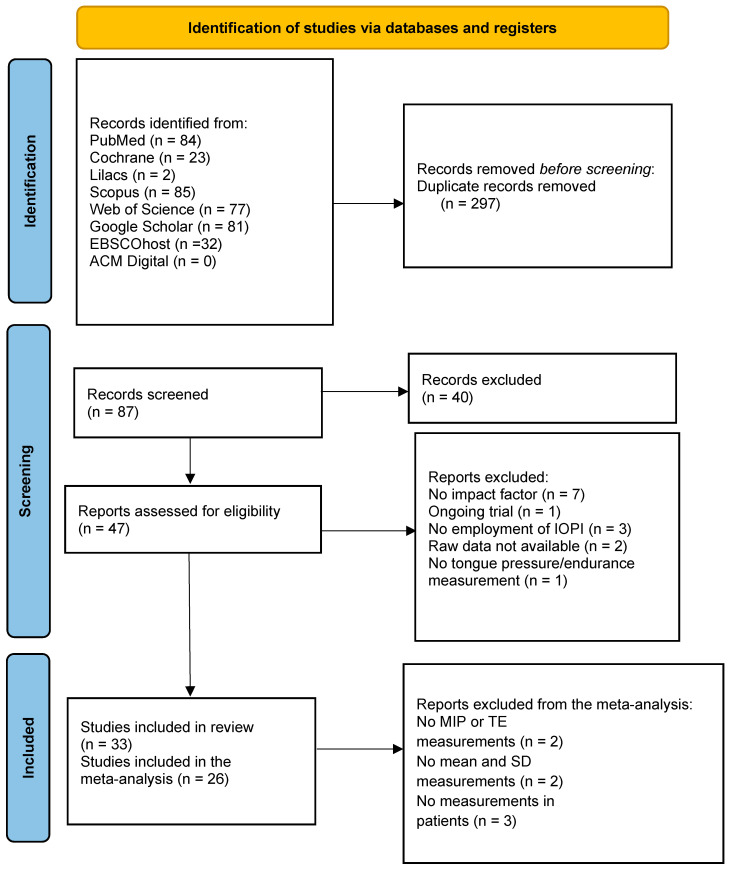
Flow chart of the search strategy.

**Figure 3 biomedicines-10-02319-f003:**
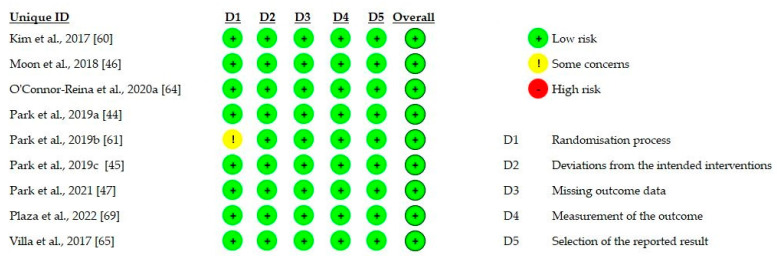
Risk of bias assessment of randomized studies using RoB 2 tool.

**Figure 4 biomedicines-10-02319-f004:**
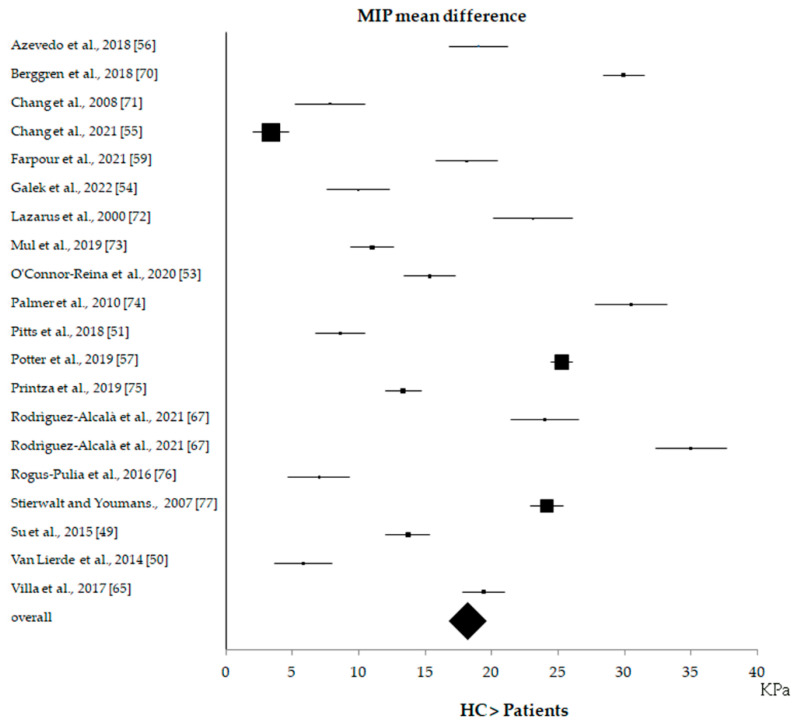
Forest plot of the maximum tongue pressure (MIP) results obtained from all 19 studies included in the case-control analysis. Rodriguez-Alcalà et al., 2021 [67] accounted for two values because the results pertained to two different groups of patients: patients affected by obstructive sleep apnea–hypopnea syndrome (OSAHS) and patients with primary snoring (PS). Villa et al., 2017 [65] had an interventional design, but MIP measurements of HC and patients were included in the case-control analysis.

**Figure 5 biomedicines-10-02319-f005:**
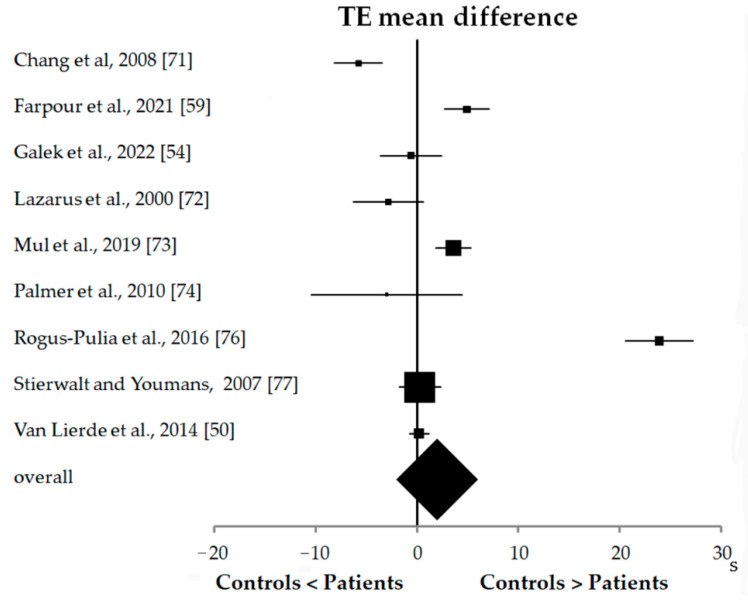
Forest plot of the tongue endurance (TE) results obtained from nine studies with case-control design.

**Figure 6 biomedicines-10-02319-f006:**
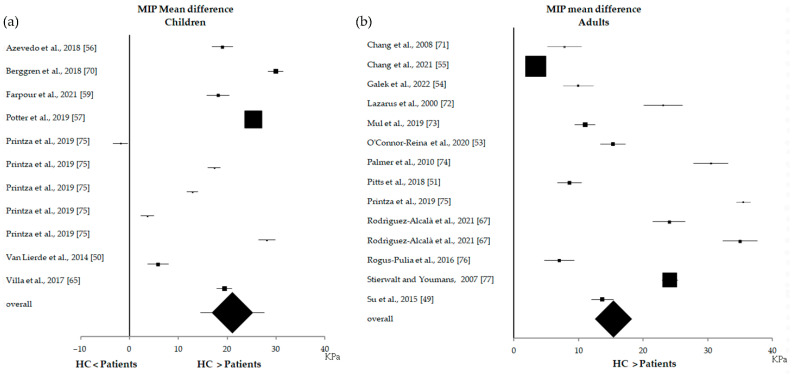
Forest plot of the MIP results from children (**a**) and adults (**b**).

**Figure 7 biomedicines-10-02319-f007:**
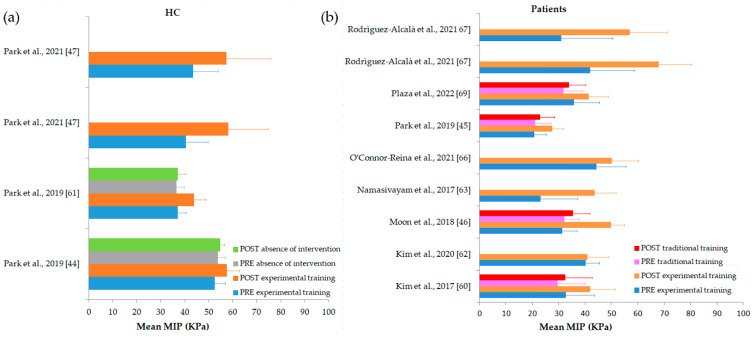
(**a**) Mean MIP values before and after experimental training and in the absence of intervention for healthy controls (HC); (**b**) before and after experimental training or traditional training for patients.

**Figure 8 biomedicines-10-02319-f008:**
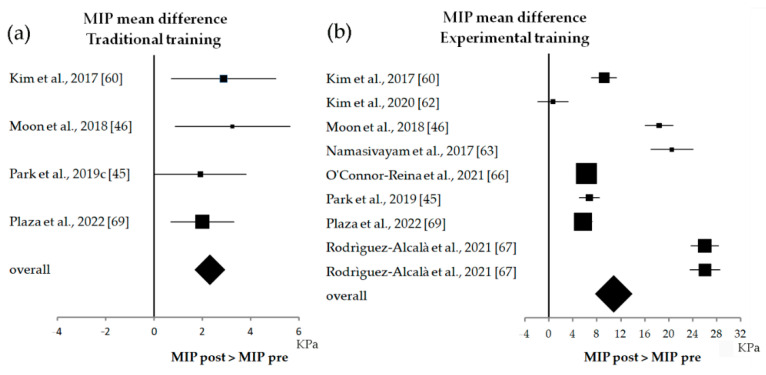
Forest plot of the MIP results for traditional training (**a**) and experimental training (**b**).

**Table 1 biomedicines-10-02319-t001:** Inclusion/exclusion criteria.

Inclusion Criteria	Exclusion Criteria
Studies in English	Studies not available in the English Language
Human studies Case-control studies, intervention studies	Preclinical/Animal studies Case reports, case series, preclinical studies, systematic reviews, meta-analyses
Studies including healthy as well as pathological populations of any age or gender Employment of IOPI	Studies referring to any diagnostic tool other than IOPI Studies published on non-impacted journals

**Table 2 biomedicines-10-02319-t002:** List of excluded studies (#14) and reasons for exclusion.

Author, Year	Reason for Exclusion
Baudelet et al., 2020 [78]	Ongoing trial
Borrman et al., 2021 [31]	No impact factor
Guzel and Tuncer, 2021 [32]	No Impact factor
Hara et al., 2014 [28]	No employment of IOPI
Hewitt et al., 2008 [38]	Raw data not available
Keskool et al., 2018 [33]	No impact factor
Kondoh et al., 2015 [29]	No employment of IOPI
Mori et al., 2017 [30]	No employment of IOPI
O’Day et al., 2005 [34]	No impact factor
Park et al., 2015 [35]	No Impact factor
Park et al., 2017 [39]	No tongue pressure or endurance measurement
Park et al., 2018 [36]	No impact factor
Potter et al., 2013 [37]	No impact factor
Steele et al., 2013 [40]	No control group and no post-intervention data

**Table 3 biomedicines-10-02319-t003:** Demographic characteristics of the studies.

Author, Year	Healthy Controls				Patients				
	n	Age Mean ± sd	Age Range	Sex (Females)	n	Age Mean ± sd	Age Range	Sex (Females)	Pathologies
Azevedo et al., 2018 [56]	20	8.3 ± 2.0	5.1–12	7	20	7.6 ± 2.2	5.1–12	7	Mouth breathing behaviour
Berggren et al., 2018 [70]	29	9.1 ± 3.1	1.3–13.9	17	41	6.8 ± 3.3	0.5–13.2	20	Congenital myotonic dystrophy
Chang et al., 2008 [71]	12	45.7	30–65	1	12	45.4	33–63	1	Nasopharyngeal carcinoma
Chang et al., 2021 [55]	336	71.6 ± 5.2	>65	196	26	75.5 ± 6.3	>65	17	Malnutrition
Farpour et al., 2021 [59]	33	10.7 ± 1.5	8.1–13	23	8	10.4 ± 1.7	8.1–13	6	Down syndrome
Galek et al., 2022 [54]	18		46–95	11	18		46–95	11	Stroke
Kim et al., 2017 [60]					ET: 18 TT: 17	ET: 62.2 ± 11.0 TT: 59.3 ± 10.2		ET: 7 TT: 9	Dysphagia following subacute stroke
Kim et al., 2020 [62]					11	75.7 ± 5.0		10	Older than 65 years, with complaints of swallowing difficulties
Lazarus et al., 2000 [72]	13	56	36–77	3	13	57	38–72	3	H and N cancer
Marim et al., 2019 [52]	23	25.5 ± 4.8		18	23	28.7 ± 6.2		19	Temporomandibular disorder
Moon et al., 2018 [46]					ET: 8 TT: 8	ET: 62 ± 4.2 TT: 63.5 ± 6.0		ET: 5 TT: 4	Stroke
Mozzanica et al., 2020 [68]					ET1: 10 ET2: 12	ET1: 8.8 ± 1.1 ET2: 19.8 ± 4.7		ET1: 6 ET2: 7	Tongue thrust
Mul et al., 2019 [73]	35	40.2 ± 13.2		18	43	52.5 ± 13.1		19	Facioscapulohumeral muscular dystrophy
Namasivayam-MacDonald et al., 2017 [63]					8	90.4	84–99	6	Seniors, with mild to moderately severe cognitive impairment in the long-term care setting
O’Connor-Reina et al., 2020a [64]					ET: 18 AI: 10	ET: 59.17 ^α^; 53.7–64.6 ^β^ AI: 63.90 ^α^; 56.4–71.38 ^β^		ET: 4 AI: 2	Sleep breathing disorders
O’Connor-Reina et al., 2020b [53]	20				35	40.6 ± 14.25		6	Sleep breathing disorders
O’Connor-Reina et al., 2021 [66]					AP: 35 NA: 19	AP: 45.9 ± 17.8 NA: 50.3	NA: 36.2–64.22	AP: 6 NA: 7	Sleep breathing disorders
Palmer et al., 2010 [74]	9	61.0	52–76	5	11	61.1	50–73	8	Oculopharyngeal muscular dystrophy
Park et al., 2019a [44]	ET:15 AI:15	ET: 24.5 ± 5.3 AI: 25.1 ± 4.2		ET: 7 AI: 8					
Park et al., 2019b [61]	ET:20 AI:20	ET: 69.5 ± 4.3 AI: 68.4 ± 3.9		ET: 10 AI: 9					
Park et al., 2019c [45]					ET: 15 TT: 15	ET: 66.5 ± 9.5 TT: 64.8 ± 11.2	ET: 51–81 TT: 53–78	ET: 6 TT: 7	Stroke
Park et al., 2021 [47]	ET1:13 ET2:13	ET1: 72.7 ± 7.3 ET2: 73.2 ± 5.7		ET1: 9 ET2: 12					
Pitts et al., 2018 [51]	28	70.7 ± 8.1		11	28	71.1 ± 8.0		11	Parkinson’s disease
Plaza et al., 2022 [69]					ET: 30 TT: 30	ET: 71.23 ± 7.64 TT: 67.46 ± 6.90		ET: 1 TT: 14	Parkinson’s disease
Potter et al., 2019 [57]	228		3.1–17	118	58		3.1–17	20	Sound and motor speech disorders
Printza et al., 2019 [75]	56		7–27		58		7–27		Muscular dystrophies
Rodrìguez-Alcalà et al., 2021 [67]	20	44 ± 6.4	18–75		ET1: 22 ET2: 21	ET1: 49.70 ± 7.90 ET2: 51.40 ± 9.30	18–75		Sleep breathing disorders
Rogus-Pulia et al., 2016 [76]	21	56.0	31–77	4	21	56.0	36–80	4	H and N cancer
Stierwalt and Youmans, 2007 [77]	200	43.8 ± 20.4	19–91	120	50	70.0 ± 13.2	44–91		Dysphagia
Su et al., 2015 [49]	36	61.5 ± 14.8		15	30	61.8 ± 15.6		8	Post-extubated patients
Van Lierde et al., 2014 [50]	25	10.6	6.7–18.2	8	25	10.7	6–17.9	8	Unilateral cleft lip and palate
Villa et al., 2017 [65]	38	7.8 ± 2.2		13	ET: 36 TT: 18	ET: 6.7 ± 2.3 TT: 6.7 ± 2.8		ET: 22 TT: 10	Sleep breathing disorders
Zanin et al., 2020 [58]	20	31.9 ± 9.3	20–51	20	19	33.2 ± 8.7	18–53	19	Sjogren’s syndrome

sd: standard deviation; ET: experimental training; TT: traditional training; H and N cancer: head and neck cancer; AI: absence of intervention; AP: adhering patients; NA: non-adhering patients. ^α^ Median value. ^β^ Interquartile range (IQR).

**Table 4 biomedicines-10-02319-t004:** Quality assessment of case-control studies, according to Newcastle–Ottawa Scale (NOS). Each study could be awarded with a maximum of one star for each item of the Selection and Exposure categories, whereas a maximum of two stars could be given for the Comparability category. Each study could be judged with a maximum of 9 stars.

Author, Year	Case Definition	Rapresentativeness	Controls Selection	Controls Definition	Comparability	Ascertainment of Exposure	Same Method	Non-Response Rate	Total Score	
Azevedo et al., 2018 [56]	*	/	*	*	**	/	*	*	7	Low
Berggren et al., 2018 [70]	*	/	*	*	*	/	*	*	5	Intermediate
Chang et al., 2008 [71]	*	/	*	*	**	/	*	*	7	Low
Chang et al., 2021 [55]	*	/	*	*	/	/	*	*	5	Intermediate
Farpour et al., 2021 [59]	*	/	*	*	**	/	*	*	7	Low
Galek et al., 2022 [54]	*	/	*	*	**	/	*	*	7	Low
Lazarus et al., 2000 [72]	*	*	*	*	**	/	*	*	7	Low
Marim et al., 2019 [52]	*	/	*	*	**	/	*	*	5	Intermediate
Mul et al. 2019 [73]	*	/	*	/	/	/	*	*	4	Intermediate
O’Connor-Reina et al., 2020b [53]	*	*	/	/	**	/	*	*	6	Intermediate
Palmer et al., 2010 [74]	*	/	/	*	*	/	*	*	5	Intermediate
Pitts et al., 2018 [51]	*	/	*	*	**	/	*	/	6	Intermediate
Potter et al., 2019 [57]	*	/	*	*	**	/	*	/	6	Intermediate
Printza et al., 2019 [75]	*	*	*	*	**	/	*	*	8	Low
Rogus-Pulia et al., 2016 [76]	*	/	*	*	**	/	*	*	7	Low
Stierwalt and Youmans, 2007 [77]	*	/	*	*	**	/	*	*	7	Low
Su et al., 2015 [49]	*	*	*	*	**	/	*	/	7	Low
Van Lierde et al., 2014 [50]	*	/	*	*	**	/	*	*	7	Low
Zanin et al., 2020 [58]	*	/	*	*	**	*	*	*	8	Low

**Table 5 biomedicines-10-02319-t005:** Risk of bias assessment of non-randomized studies using ROBINS-I tool.

Author, Year	Confounding	Selection of Participants	Classification of Interventions	Deviations from Intended Interventions	Missing Data	Measurement of Outcomes	Selection of the Reported Result	Overall Bias
Kim et al., 2020 [62]	Low	Low	Low	Low	Low	Low	Low	Low
Mozzanica et al., 2020 [68]	Low	Low	Low	Low	Low	Low	Low	Low
Namasivayam-MacDonald et al., 2017 [63]	Low	Moderate	Low	Low	Low	Low	Low	Moderate
O’Connor-Reina et al., 2021 [66]	Low	Low	Low	Low	Low	Low	Low	Low
Rodriguez-Alcalà et al., 2021 [67]	Low	Low	Low	Low	Low	Low	Low	Low

**Table 6 biomedicines-10-02319-t006:** IOPI measures for the case-control studies.

Author, Year	Primary Outcomes		Secondary Outcomes					
	MIP (kPa) Mean ± sd		LSP (kPa) Mean ± sd		TE (s) Mean ± sd		TPS (kPa) Mean ± sd	
	HC	Patients	HC	Patients	HC	Patients	HC	Patients
Azevedo et al., 2018 [56]	51.4	32.4						
Berggren et al., 2018 [70]	41.81	11.88						
Chang et al., 2008 [71]	64.5 ± 12.57	56.67 ± 9.35			18.75 ± 6.22	24.58 ± 10.72		
Chang et al., 2021 [55]	38.2 ± 14.01	34.84 ± 11.57						
Farpour et al., 2021 [59]	31.43 ± 15.39	13.29 ± 7.46			11.5 ± 7.66	6.6 ± 5.12		
Galek et al., 2022 [54]	42 ± 11.16	32.05 ± 14.66	30.33 ± 11.89	28.22 ± 13.36	43.77 ± 20.24	44.38 ± 22.25		
Lazarus et al., 2000 [72]	60.15 ± 3.68	37.05 ± 4.56			37.77	40.62 ± 7.8		
Marim et al., 2019 [52]			30.3 ± 12	23.1 ± 9			47.2 ± 17.2	35.3 ± 17
Mul et al. 2019 [73]	61.45 ± 10.7	50.45 ± 15.65			29 ± 14.7	25.4 ± 14.8		
O’Connor-Reina et al., 2020b [53]	59.34 ± 12.3	44.01 ± 12.2						
Palmer et al., 2010 [74]	57.4 ± 10.4	26.9 ± 7.8	22.3 ± 8.9	10.4 ± 2.4	119 ± 63	122 ± 43		
Pitts et al., 2018 [51]	54.5 ± 10.6	45.9 ± 15	18.7 ± 10	17.55 ± 9.8				
Potter et al., 2019 [57]	51.53 ± 7.69	26.24 ± 7.75						
Printza et al., 2019 [75]	49.86 ± 13.7	36.51 ± 14.68						
Rogus-Pulia et al., 2016 [76]	58 ± 15.8	51 ± 13.7			68.9 ± 43.8	45 ± 16.4		
Stierwalt and Youmans, 2007 [77]	59.78 ± 13.73	35.64			39.62	39.34 ± 43.28		
Su et al., 2015 [49]	47.4 ± 9.6	33.7 ± 13.67						
Van Lierde et al., 2014 [50]	43 ± 14.8	37.2 ± 15.3			3.9 ± 3.7	3.7 ± 2.3		
Zanin et al., 2020 [58]							41.3 ± 14.4	24.2 ± 12.8

Drop out: 11 [49], 1 [75]. sd: standard deviation; HC: healthy controls; P: patients; MIP: maximum tongue pressure; LSP: lingual swallowing pressure; TE: tongue endurance; TPS: tongue protrusion strength.

**Table 7 biomedicines-10-02319-t007:** IOPI measures for the intervention studies.

Author, Year	Intervention		Follow Up (Weeks)	Primary Outcomes				Secondary Outcomes							
				MIP (kPa)				LSP (kPa)				TE (s)			
	HC	Patients		HC		Patients		HC		Patients		HC		Patients	
				Pre- Treatment	Post- Treatment	Pre- Treatment	Post- Treatment	Pre- Treatment	Post- Treatment	Pre- Treatment	Post-Treatment	Pre-Treatment	Post-Treatment	Pre-Treatment	Post-Treatment
Kim et al., 2017 [60]		ET: Tongue-to-palate resistance training + traditional dysphagia therapy TT: Traditional dysphagia therapy	4			ET: 32.67 ± 10.78 TT: 29.65 ± 10.41	ET: 41.89 ± 9.54 TT: 32.53 ± 10.17								
Kim et al., 2020 [62]		mHealth app therapy program	12			ET: 40.30 ± 5.10	TT: 41 ± 8			ET: 18.30 ± 6.50	ET: 27.40 ± 9.40				
Moon et al., 2018 [46]		ET: Tongue pressure strength and accuracy training TT: Traditional dysphagia therapy	8			ET: 31.38 ± 5.68 TT: 32.25 ± 5.37	ET: 49.75 ± 5.26 TT: 35.50 ± 6.35								
Mozzanica et al., 2020 [68]		Oral myofunctional therapy	10			ET1: 31 ^α^; 27–42.5 ^β^ ET2: 23.50 ^α^; 19.5–33.2 ^β^	ET1: 47 ^α^; 39.5–49.5 ^β^ ET2: 41.50 ^α^; 36.75–45.75 ^β^								
Namasivayam-MacDonald et al., 2017 [63]		Tongue strengthening training				ET: 23.10 ± 14.08	ET: 43.62 ± 8.10								
O’Connor-Reina et al., 2020a [64]		mHealth app myofunctional therapy	12			ET: 40.26 ^α^; 35.32–45.2 ^β^ AI: 42 ^α^; 32.67–51.33 ^β^	ET: 59.06 ^α^; 54.74–64 ^β^ AI: 44.2 ^α^; 34.1–54.2 ^β^								
O’Connor-Reina et al., 2021 [66]		AirwayGym app myofunctional therapy	12			AP: 44.40 ± 11.08 NA: 51.30 ± 11.40	AP: 50.66 ± 10.20 NA: 51.10 ± 11.17								
Park et al., 2019a [44]	Lingual strength training		6	ET: 52.50 ± 4.44 AI: 53.81 ± 3.01	ET: 57.66 ± 5.21 AI: 54.72 ± 1.95										
Park et al., 2019b [61]	Tongue strengthening exercise			ET: 37.08 ± 3.50 AI: 36.55 ± 3.29	ET: 43.92 ± 4.88 AI: 37.09 ± 3.36										
Park et al., 2019c [45]		ET: Effortful swallowing training + conventional dysphagia treatment TT: Saliva swallowing + conventional dysphagia treatment	4			ET: 20.83 ± 4.32 TT: 21.16 ± 5.78	ET: 27.58 ± 4.27 TT: 23.08 ± 5.42								
Park et al., 2021 [47]	ET1: Tongue resistance exercise ET2: Tongue isometric exercise		4	ET1: 40.50 ± 9.20 ET2: 43.50 ± 10.40	ET1: 58.10 ± 16.70 ET2: 57.50 ± 18.30			ET1: 26.10 ± 12.40 ET2: 31.30 ± 12.60	ET1: 38 ± 22.70 ET2: 32.50 ± 22.70						
Plaza et al., 2022 [69]		ET: Tongue isometric pressure exercises + traditional tongue therapy TT: Traditional tongue therapy	12			ET: 35.80 ± 9.58 TT: 31.90 ± 6.96	ET: 41.50 ± 7.39 TT: 33.90 ± 6.38								
Rodriguez-Alcalà et al., 2021 [67]		AirwayGym app myofunctional therapy	12	66 ± 18.20		ET1: 42.00 ± 16.70 ET2: 31.00 ± 19.50	ET1: 68.00 ± 12.40 ET2: 57.00 ± 14.20								
Villa et al., 2017 [65]		ET: Myofunctional therapy + nasal washing TT: nasal washing	8	51.30 ± 13.60		ET: 31.90 ± 10.70 TT: 32.40 ± 9.40						15.80 ± 7.20	ET: 28.10 ± 8.90 TT: 23.30 ± 5.90		

Drop out: 6 [60], 3 [62], 3 [46], 12 [64], 6 [45], 3 [47], 7 [67]. HC: healthy controls; P: patients; ET: experimental training; TT: traditional training; AI: absence of intervention; AP: adhering patients; NA: non-adhering patients; MIP: maximum tongue pressure; LSP: lingual swallowing pressure; TE: tongue endurance. ^α^ Median value. ^β^ Interquartile range (IQR).

## Data Availability

Research data not shared.

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
