# Peer review of "Quantitative Measurement of Swallowing Performance Using Iowa Oral Performance Instrument: A Systematic Review and Meta-Analysis"

_biomedicines, 2022, doi:10.3390/biomedicines10092319_

Round 1

Reviewer 1 Report

The authors of the article submitted for review are Raffaella Franciotti, Erica di Maria, Michele D’ Attilio, Giuseppe Aprile, Federica Giula Cosentino and Vittoria Perrotti. The main focus of their publications are issues concerning the use of artificial intelligence in different areas of medicine. 

In this manuscript the authors have made an attempt to assess the effectiveness of a relatively new device called IOPI used in detecting swallowing disorders, pathologies and conditions that imply dysphagia. 

The publication is a literature review relating to research of the IOPI device and its functionality. The authors have performed a systematic literature review using the following databases: PubMed, Cochrane, Lilacs and Scopus. What is incomprehensible to me, and needs to be explained, is the fact that other similar databases have been totally omitted in the research process, including: ACM Digital, Base, EBSCOhost, Google Scholar or Web of Science. They are equally important and in my opinion the authors should not limit themselves to only a few most popular sources. 

What raises doubts as well is that research coming from periodicals that do not have the IF (impact factor) has been put into the exclusion criteria. This matter especially requires a detailed explanation. I am firmly convinced, having personal experience in that area, that there are valuable scientific articles being published in those periodicals that do not have the said impact factor for one reason or another.  

The authors put a strong focus on comparing the presented studies, whilst almost entirely overlooking the description of the device itself. I highly recommend adding a whole paragraph dedicated to illustrating the exact way IOPI works (with pictures) that would allow any potential readers to more accurately understand its application and principles. 

A positive aspect of this manuscript is the inclusion of numerous charts, graphs and diagrams, that undoubtedly allow for a deeper analysis of the subject and its better perception. 

The structural and linguistic aspects of this work do not give rise to any objections. 

Author Response

REVIEWER 1

Comment of the reviewer: The authors of the article submitted for review are Raffaella Franciotti, Erica di Maria, Michele D’ Attilio, Giuseppe Aprile, Federica Giula Cosentino and Vittoria Perrotti. The main focus of their publications are issues concerning the use of artificial intelligence in different areas of medicine. In this manuscript the authors have made an attempt to assess the effectiveness of a relatively new device called IOPI used in detecting swallowing disorders, pathologies and conditions that imply dysphagia. 

The publication is a literature review relating to research of the IOPI device and its functionality. The authors have performed a systematic literature review using the following databases: PubMed, Cochrane, Lilacs and Scopus. What is incomprehensible to me, and needs to be explained, is the fact that other similar databases have been totally omitted in the research process, including: ACM Digital, Base, EBSCOhost, Google Scholar or Web of Science. They are equally important and in my opinion the authors should not limit themselves to only a few most popular sources. 

Our response: Thank you for your comment. We checked the search, and we find out that by mistake the search on Web of Science was actually conducted as mentioned in line 142 of the section search strategy of the materials and methods, but the results were not presented in the flow chart. Therefore, the flow chart was updated with those results as well as with the results of the other databases you suggested (ACM Digital, Base, EBSCOhost, Google Scholar). In addition, the new databases searched were added also in the search strategy paragraph of the materials and methods section.

Comment of the reviewer: What raises doubts as well is that research coming from periodicals that do not have the IF (impact factor) has been put into the exclusion criteria. This matter especially requires a detailed explanation. I am firmly convinced, having personal experience in that area, that there are valuable scientific articles being published in those periodicals that do not have the said impact factor for one reason or another.  

Our response: We thank the reviewer for bringing this issue to our attention. We understand this point of view which we agree, however, the protocol of the present review has been registered on the international prospective register of systematic reviews PROSPERO (identification number CRD42022297506), thus our criteria have been approved.

Comment of the reviewer: The authors put a strong focus on comparing the presented studies, whilst almost entirely overlooking the description of the device itself. I highly recommend adding a whole paragraph dedicated to illustrating the exact way IOPI works (with pictures) that would allow any potential readers to more accurately understand its application and principles. 

Our response: We completely agree with the Reviewer. Thus, a paragraph as well as 3 figures compiled into a panel (Fig. 1 a-c) have been added in the introduction (lines 88-133) section to allow the readers a more detailed understanding of how IOPI is made and how it works. A copyright 

permission to use the above mentioned images - released by IOPI - has been uploaded as supplementary material.

Comment of the reviewer: A positive aspect of this manuscript is the inclusion of numerous charts, graphs and diagrams, that undoubtedly allow for a deeper analysis of the subject and its better perception. 

Our response: We thank the reviewer and the editor for the positive and encouraging comment.

Comment of the reviewer: The structural and linguistic aspects of this work do not give rise to any objections. 

Our response: Thank you.

Reviewer 2 Report

Few corrections are required

(The Authors must see my remarks)

Author Response

REVIEWER 2

Comment of the reviewer: Reference(s)?

Our response: The eligibility criteria can be adopted from another review, or it can be newly developed by the authors. For this paper the authors selected ad hoc eligibility criteria, thus References for eligibility criteria, inclusion and exclusion criteria were not provided. Moreover, the protocol has been registered on the international prospective register of systematic reviews PROSPERO (identification number CRD42022297506), thus these criteria have already been approved.  

Comment of the reviewer: Reference(s)?

Our response: References numbers of the excluded studies (i.e. studies reporting any other diagnostic tool and not referring to IOPI measures as well as the ones published on journals without an impact factor and not peer-reviewed) have now been added at lines 176-181. 

Comment of the reviewer: No questions in Scientific articles....

Our response: Thank you for your suggestion. The paragraphs have been modified reporting first the PICO (Participants, Intervention, Comparison, Outcomes) Model for Clinical Questions and rephrasing our hypotheses to avoid direct questions (lines 184-196).

Comment of the reviewer: Reduce this Section and state the main outcomes only....

Our response: Thank you for your comment. The conclusion section has now been shortened to obtain a more schematic synthesis of the results (lines 624-641). 

Round 2

Reviewer 1 Report

The authors have addressed the reviewer's comments fully and with great care. The previously existing concerns have been thoroughly explained and all the doubts dispeled. Therefore, I believe that the manuscript is eligible for acceptance after this correction and can be submitted for publishing.